# The Interaction between Congruency and Numerical Ratio Effects in the Nonsymbolic Comparison Test

**DOI:** 10.3390/bs13120983

**Published:** 2023-11-28

**Authors:** Yulia Kuzmina, Julia Marakshina, Marina Lobaskova, Ilya Zakharov, Tatiana Tikhomirova, Sergey Malykh

**Affiliations:** Psychological Institute of Russian Academy of Education, 125009 Moscow, Russia; jkuzmina@hse.ru (Y.K.); retalika@yandex.ru (J.M.); lobaskova@raop.ru (M.L.); iliazaharov@pirao.ru (I.Z.); tikho@mail.ru (T.T.)

**Keywords:** Approximate Number System, number sense, visual cues, congruency effect, numerical ratio effect

## Abstract

The nonsymbolic comparison task is used to investigate the precision of the Approximate Number Sense, the ability to process discrete numerosity without counting and symbols. There is an ongoing debate regarding the extent to which the ANS is influenced by the processing of non-numerical visual cues. To address this question, we assessed the congruency effect in a nonsymbolic comparison task, examining its variability across different stimulus presentation formats and numerical proportions. Additionally, we examined the variability of the numerical ratio effect with the format and congruency. Utilizing generalized linear mixed-effects models with a sample of 290 students (89% female, mean age 19.33 years), we estimated the congruency effect and numerical ratio effect for separated and intermixed formats of stimulus presentation, and for small and large numerical proportions. The findings indicated that the congruency effect increased in large numerical proportion conditions, but this pattern was observed only in the separated format. In the intermixed format, the congruency effect was insignificant for both types of numerical proportion. Notably, the numerical ratio effect varied for congruent and incongruent trials in different formats. The results may suggest that the processing of visual non-numerical parameters may be crucial when numerosity processing becomes noisier, specifically when numerical proportion becomes larger. The implications of these findings for refining the ANS theory are discussed.

## 1. Introduction

Numerous studies indicate that humans and animals are capable of processing quantitative information without symbols, specifically, they can compare arrays of objects and detect the largest one, detecting changes in numerosities, or establishing similarities between quantities without [1,2,3]. This ability is usually referred to as Approximate Number Sense (ANS) [4,5]. The theory of ANS postulates that there exists a separate innate system that provides rapid but imprecise processing of quantitative information and that numerosity can be processed directly [4,6]. In support of this theory, numerous studies have demonstrated that the processing of numerical information occurs at an early stage of visual processing and is similar to the processing of other non-numerical visual properties, like shape, color, or size [6,7,8,9].

To investigate the ANS precision, scientists use various types of tasks, with one of the most popular being the nonsymbolic comparison task [10,11]. In this task, participants are required to compare two arrays of geometric figures or other objects presented for a very short time (200–1500 ms) and select which array contains more objects. The use of the nonsymbolic comparison task has enabled researchers to identify different aspects of processing numerosity without using symbols. One of the main features is the numerical ratio effect (NRE) or numerical distance effect (NDE) [12,13]. The NRE (NDE) is manifested as an increase in reaction time and/or a decrease in accuracy when comparing arrays that are closer in numerosity or, in other words, have a smaller numerical distance [14,15,16].

One hypothesis regarding the existence of the NRE is based on the theory of the mental number line, a form of representing numbers in the brain [17,18]. It is believed that perceived numerosities are located along this analog mental number line, which is oriented from left to right (in cultures with left-to-right writing direction) [19,20]. Each number corresponds to a population of neurons that fire while processing a certain number (in any format, symbolic, nonsymbolic, or verbal), and the activation of these neurons can be represented as Gaussian curves [21,22]. Consequently, when numbers are located closer to each other on the mental number line, they have more overlapping activated neurons, and number representation or discrimination can be noisier. However, some authors have suggested that the NRE does not imply the overlapping internal representations for two quantities and instead may arise from relative number word frequency (for digit comparison) and reflect response selection processes [23,24]. It should also be noted that although the existence of the NRE is supported by many studies, some scholars have noted that it has low reliability and should not be used as an indicator of individual precision of the ANS [25,26].

The second feature of nonsymbolic comparison is its dependence on the estimation and comparison of non-numerical continuous visual cues, such as cumulative area, density, convex hull, etc. [27,28,29]. This dependence is manifested in the congruency effect: higher accuracy in nonsymbolic comparison when visual cues positively correlate with numerosity (congruent trials) compared to trials where visual cues negatively correlate with numerosity (incongruent trials) [30,31]. For example, in congruent trials, the array containing more objects has a larger cumulative area or convex hull, while in incongruent trials, the array with more objects has a smaller cumulative area.

Considering the existence of the congruency effect, several scientists doubted the existence of the separate system of numerosity processing and suggested the Sensory Integration theory, which opposed the ANS theory [32]. They believe that the congruency effect in nonsymbolic comparison tasks indicates that the only way to process discrete numerosity is to estimate and compare non-numerical continuous visual characteristics [30,32]. Various visual cues are processed and weighted to make a final decision about numerosity [32]. It has been suggested that instead of the Approximate Number System, there is a common system for processing continuous magnitudes and discrete numerosity (the General Magnitude System or the Approximate Magnitude System) [28,32]. This system forms the basis for the development of processing numerosity in any format, processing time, and visual properties (e.g., lengths, areas).

Another view on the congruency effect suggests that the estimation of visual cues affects the estimation of numerosity, but processing numerosity is a holistic process that combines the estimation of numerosity and visual cues [33]. In this case, the congruency effect arises due to the insufficient inhibition of irrelevant non-numerical visual information [34,35]. Numerosity can be processed separately, but non-numerical visual cues are more salient, making it difficult to inhibit them.

Several findings have been obtained that confirmed the interplay between the processing of numerosity and non-numerical visual cues. First, it has been shown that participants are able to process several visual parameters. In particular, scholars have manipulated several visual features and identified partial (in)congruency and full (in)congruency conditions [30,36]. In partial (in)congruency conditions, arrays were (in)congruent in one of the controlled visual features (e.g., cumulative area), while in fully (in)congruency conditions, arrays were (in)congruent on all controlled visual properties [30]. It was shown that in fully incongruent conditions, accuracy was significantly lower than in partially incongruent conditions and that the accuracy in fully incongruent conditions was lower 50% [30,31].

Second, it has been shown that some visual parameters skew numerosity estimation more significantly than others [7,36,37]. Two types of visual cues have been identified: extrinsic and intrinsic [38]. Extrinsic features, such as convex hull or density, provide information about arrays of objects as a whole. Intrinsic features (such as the cumulative area or size of each object) are based on the estimation of each object and the calculation of the characteristics of arrays based on these individual characteristics. It was shown that extrinsic features (e.g., convex hull) had a greater effect than intrinsic features [37,38]. Moreover, it was shown that not only the area within the convex hull but also the convex hull’s shape affects numerosity judgment. In general, numerous findings supported the Sensory Integration theory and demonstrated that the estimation of numerosity is biased due to the processing of non-numerical characteristics.

At the same time, several findings regarding the congruency effect did not support the Sensory Integration theory. Particularly, it has been shown that the congruency effect varies depending on the format of stimulus presentation [39,40]. The congruency effect is significantly smaller (or absent) when arrays are presented in a spatially intermixed format than in a spatially separated format [39]. Additionally, in incongruent trials, the accuracy of nonsymbolic comparison was higher in the intermixed format than in the separated format.

The differences between formats can be explained by differences in the processing of visual cues in intermixed and separated formats. Particularly, in the intermixed format, the estimation of two main visual properties (cumulative area and convex hull) was difficult. Despite this, the estimation of numerosity remained accurate in this format. Thus, accurate numerosity estimation was possible even when the estimation of non-numerical visual cues was impeded. This finding supports the ANS theory rather than the Sensory Integration theory [39].

Despite intensive investigation of numerosity processing, little is known about how two main effects of numerosity processing, namely NRE and congruency effect, interact with each other. While NRE reflected the “noisiness” of numerosity estimation per se, the congruency effect reflected “noisiness” of the numerosity estimation due to the estimation of non-numerical visual cues. Previously, it was shown that the numerical ratio and ratio between areas both affected the accuracy of nonsymbolic comparison [41]. However, it is not clear how the bias in the estimation of numerosity due to the processing of non-numerical features varied depending on the “noiseness” of numerical processing.

### Current Study

In this study, we are going to investigate the interaction between congruency effect and NRE in different formats of stimulus presentation, assuming that different formats of presentation imply different processing of non-numerical visual cues [39]. Previous studies on the modulation of the congruency effect in different formats had several limitations [39,40]. First, a narrow range of numerical ratios between two arrays was used, that restricted obtained findings. Second, the congruency effect was separately assessed for two visual cues, namely convex hull and cumulative area. Since accuracy can significantly decrease in fully incongruent conditions, where arrays are incongruent on both visual cues, it is crucial to explore the variability in the congruency effect under full incongruence [31].

The current study has several aims. First, we aimed to estimate whether the congruency effect would vary in different formats of stimulus presentation in the case of full incongruency. Second, we aimed to test whether the NRE varied depending on the format of presentation. Finally, we aimed to estimate the variability of the congruency effect for different numerical ratios and formats jointly.

To fulfill these aims, we used a nonsymbolic comparison test with two formats of stimulus presentation: spatially separated with homogeneous figures (separated/homogeneous) and spatially intermixed with heterogeneous figures (intermixed/heterogeneous). For each format, we included congruent and incongruent trials and manipulated the numerical proportion for each type of trial.

We formulated the corresponding hypotheses regarding our aims:(1)The congruency effect would be smaller in the intermixed/heterogeneous format than in the separated/homogenous format even in the case of full incongruency. We hypothesized that the difference between formats might be explained by the difficulties in estimating visual cues. In the intermixed/heterogeneous format, the comparison of convex hull and cumulative areas might be distorted. From this perspective, in intermixed/heterogeneous conditions, participants may not be able to accurately estimate visual cues and would have to rely mostly on the estimation of numerical features;(2)The NRE would be higher in the intermixed/heterogeneous format than in the separated/homogenous format. In a previous study, it was suggested that visual and numerical features of compared arrays can be processed in parallel [39]. Their interrelationships may vary depending on how easy and accessible it is to evaluate and compare non-numerical visual parameters. We hypothesized that if in intermixed/heterogeneous conditions, the estimation of visual cues is impeded, numerical features would be more salient than non-numerical visual features. Therefore, the NRE would increase in the intermixed format;(3)The congruency effect would increase with a larger numerical proportion, and this dependency would be more pronounced in the separated format. This hypothesis is based on the assumption that the estimation of visual and numerical information can be processed in parallel, but in incongruent trials, these processes come into conflict. Consequently, if the estimation of numerical information becomes more noisy due to an increase in numerical proportion, the effect of visual cues and the congruency effect might increase. The congruency effect would be higher when the estimation of visual parameters is easier than the estimation of numerical information. In such a case, the congruency effect would be more pronounced in the separated format and when the numerical proportion between the two sets is high.

## 2. Materials and Methods

### 2.1. Sample

Data were collected from a sample of adults via the online platform (pavlovia.org (accessed on 17 November 2020)). Participants were recruited from universities in two cities in Russia (Ekaterinburg and Izhevsk). Participation was voluntary, and participants were compensated for their participation through the Pavlovia platform. Initially, 344 participants began the test, but not all of them completed it. We analyzed data from participants who completed the test and reported having normal or corrected vision with no color vision disorders. The final sample consisted of 290 participants, of whom 89% were female. The mean age was 19.33 years (SD = 1.64, range 17–27).

### 2.2. Instrument and Procedure

#### Nonsymbolic Comparison Test

During the test, two sets of red and green figures were shown to the respondents for a short duration. They were required to select which array (red or green) contained more figures by pressing the button with the corresponding letter denoting the color: “r”—if there were more red figures, “g”—if there were more green figures. Each screen displaying two sets of figures was shown for 400 ms, after which the image disappeared and was replaced by a screen with a reminder: “Press “r” if there are more red figures, press “g” if there are more green figures”. After making a selection and pressing a key, a fixation cross was presented for 400 ms, followed by the next screen displaying two sets of figures.

The test design incorporated three dimensions: congruency, format of presentation, and numerical ratio. In half of the trials, the stimulus could be incongruent on two visual cues: convex hull (the minimal perimeter that included all figures of the same color) and cumulative areas (the sum of the areas of all figures of the same color). In the congruent condition, the array that contained more figures had a larger cumulative area and convex hull, while in the incongruent condition, the array containing more dots had a smaller cumulative area and convex hull.

There were two formats of stimulus presentation: spatially separated with homogeneous figures (separated/homogeneous format) and spatially intermixed (intermixed) with heterogeneous figures (intermixed/heterogeneous format). In the separated/homogenous format, participants compared red circles with green circles (Figure 1A). In the intermixed/heterogeneous format, participants compared red circles with green triangles (or green circles with red triangles).

The choice of these formats was based on the results of previous studies that demonstrated that the separated/homogeneous format produced a large congruency effect, while in the intermixed/heterogeneous format, the congruency effect disappeared [39,40]. In the previously mentioned studies, items might be congruent on one visual parameter, while another was constrained to be equal for two arrays. In the current study, stimuli were congruent or incongruent on two parameters.

Regarding the numerical proportion, two types of numerical proportions were included: small and large. For a small proportion, the ratio of a smaller quantity to a larger one varied from 0.47 to 0.53 (a smaller quantity divided by a larger one), while for a large proportion, it ranged from 0.72 to 0.77. In half of the trials in each condition, the green array contained more figures, and in the other half, the red array contained more figures.

The intersection of these three dimensions resulted in eight conditions, each with twenty-four trials. A brief description of each condition can be found in Table 1.

It is evident from Table 1, that in the “small proportion” condition, four numerical proportions were used: 9:18, 9:19, 16:8, and 19:10. Consequently, the numerical size, defined as the sum of objects in two arrays, ranged from 24 to 29. In the “large proportion” condition, the following numerical proportions were utilized: 9:12, 13:18, 13:10, and 16:12. Thus, the numerical size varied from 21 to 31.

We also added one additional condition to estimate whether any differences existed in the comparison of circles and triangles in the separated/homogeneous format. This condition was identical to the first condition (separated/homogenous, small proportion, congruent), but instead of circles, participants compared two arrays of triangles. In total, the test contained 216 items. Items from different conditions were presented in a random order, which was the same for all participants.

### 2.3. Statistical Approach

In the first step, we calculated the proportion of correct answers on the test and in each condition. We also compared whether any differences existed between comparing triangles and comparing circles.

Next, we applied generalized linear mixed-effects models (GLMMs) to estimate the effects of congruency, format of presentation, and numerical proportion, as well as the interactions between them. Mixed-effects models allow us to separate between-individual variance and within-individual variance in item answers. GLMMs modeled the probability of a correct answer and estimated the effect of different predictors on the item level (congruency, format, and proportion) or at the subject level. GLMMs also allow us to estimate between-subject differences in the effects of item-level variables via random slope models [42].

We started with a baseline model (without predictors). Next, we included the variables “format” (0—separated, 1—intermixed), “congruency” (0—congruent, 1—incongruent), and “proportion” (0—small, 1—large). Then, we included interactions between variables: format and congruency, format and proportion, and proportion and congruency. Finally, we estimated the three-way interaction between format, proportion, and congruency. Each subsequent model was compared with the previous model via a likelihood ratio test (LR test). If this test was significant, it indicated that the model with more parameters fitted the data better than the model with fewer parameters.

The analysis was conducted using Stata 17.0 Software [43].

## 3. Results

### 3.1. Descriptive Statistics

The proportions of correct answers for each condition are shown in Table 2.

Descriptive statistics revealed that in the whole test and in each condition, the accuracy was higher than 0.50. The mean accuracy in congruent trials was higher than that in incongruent trials. Additionally, participants were more precise when comparing arrays with a small numerical proportion than with a large numerical proportion.

We have also assessed the average accuracy in each condition separately (Table 3).

These results revealed that accuracy was lower in large proportion than in small proportion for the separate and intermixed format, for the congruent and incongruent trials. At the same time, the accuracy in the incongruent trial was lower than in the congruent in the separate format only.

We also tested whether there was any difference between the results of comparing circles and triangles in the separated/homogeneous format. We calculated the average accuracy for the comparison of circles and triangles. The results revealed that there were no differences in accuracy between the comparison of circles and triangles in the separated/homogeneous format.

### 3.2. Results of GLMM

To estimate the significance of differences between conditions and interactions between effects, the GLMM was applied. The results of the analysis are presented in Table 4.

The results of the baseline model revealed that the intraclass correlation coefficient (ICC), the ratio of between-individual variance to the total variance, was equal to 0.09. Hence, the majority of the variance in the probability of a correct answer was related to between-items variance.

The results of Model 1 indicated that when controlling for congruency and numerical proportion, the probability of a correct answer in the intermixed format was higher than that in the separated format. The probability of a correct answer was lower in incongruent trials than in congruent trials and in the large proportion condition than in the small proportion condition.

The results of Model 2 revealed that two interactions were significant: the interaction between the format and congruency and the interaction between congruency and numerical proportion. The regression coefficient for the interaction between format and congruency was positive. Assuming that the coefficient of “incongruency” was negative, the probability of a correct answer for incongruent trials was lower, but in the separated format only. In the intermixed format, the probability of a correct answer in the incongruent trials increased, so the difference between congruent and incongruent trials became insignificant. On the other hand, this interaction revealed that the difference between formats varied for congruent and incongruent trials. Participants were more accurate in the intermixed format than in the separate format, but only in incongruent trials.

The probabilities of correct answers for congruent and incongruent trials for intermixed and separated formats are presented in Figure 2.

The coefficient for the interaction between congruency and numerical proportion was negative, indicating that the congruency effect is more salient in the large numerical proportion condition (Figure 3).

Regarding our first hypothesis, the results revealed that the congruency effect was higher in the separated format. However, the results did not confirm the second hypothesis—the numerical proportion effect was not higher in the intermixed format than in the separated format. The results of the post-estimation analysis are presented in Table 5.

Model 3 with three-way interactions demonstrated that each interaction was significant. To detect the congruency effect in different formats and numerical proportion conditions, we conducted a post-estimation analysis and calculated the congruency effect in each format for large and small proportions (Table 6).

These results demonstrated that the congruency effect (the difference between congruent and incongruent trials) was significant only in the separated format and that the congruency effect was larger in the large proportion condition than in the small proportion condition.

In general, the probability of a correct answer was higher in congruent trials than in incongruent trials, but this was observed only in the separated format. Additionally, the difference between congruent and incongruent trials was more pronounced in the large proportion conditions (Figure 4).

We also evaluated whether the NRE varied depending on congruency and format. The analysis revealed that in the separated format, the NRE was higher in incongruent trials than in congruent trials. In contrast, in the intermixed format, the NRE was larger in congruent trials than in incongruent ones (Table 7).

## 4. Discussion

Approximate Number Sense (ANS) is the ability to process numerical information without using symbols. There are two concurrent theories regarding the ANS and its relationships with visual cues. ANS theory implies that numerosity is a separate visual property that can be processed independently of other visual features [44]. Another theory, known as the Sensory Integration theory, suggests that the processing of numerosity is based on the processing of multiple visual features and that numerical information cannot be perceived directly [32]. The existence of the congruency effect can confirm the close relationship between the estimation of non-numerical visual cues and numerosity.

In this study, we aimed to expand previous studies and estimate how the congruency effect varied depending on the numerical proportion and the format of stimulus presentation. We also aimed to assess whether the NRE varied depending on the format of stimulus presentation. We used the nonsymbolic comparison test with two formats: spatially intermixed with heterogeneous geometric shapes (intermixed/heterogeneous) and spatially separated with the same geometric shapes (separate/homogenous). The trials could be congruent or incongruent for two visual parameters simultaneously, convex hull and cumulative area. The congruency effect was estimated as the differences between congruent and incongruent trials. The NRE was estimated as the difference between the small numerical proportion (0.47–0.53) and the large numerical proportion (0.72–0.77).

Two hypotheses regarding the congruency effect were suggested. The first hypothesis was confirmed, and the congruency effect was higher in the separated format than in the intermixed format. The findings obtained in previous studies were confirmed for the case of incongruency in two visual cues.

Next, we tested the hypothesis that the congruency effect would be higher when the numerical ratio was higher and that this variability would be more salient in the separated format. Our analysis revealed that the congruency effect was significant only in the separated format. In this format, the congruency effect was higher in the large proportion condition than in the small proportion condition. This might indicate that the estimation of visual cues can serve as a supportive process for numerosity estimation. The evaluation of visual parameters may be required if the evaluation of quantitative information becomes more complex. In part, these results are in line with a recent study by Kang and Ratcliff (2020), who demonstrated that the effects of the non-numerical feature variables were moderated by the numerical properties (number of comparing objects) [45].

Our analysis also revealed that the NRE does not vary in different formats of stimulus presentation if it is estimated for congruent and incongruent trials together. However, the NRE varied depending on both congruency and format. The NRE was larger in incongruent trials than in congruent trials, but only in the separated format. One possible source of decreased accuracy in incongruent trials is the lack of inhibition ability [35]. In this case, it is possible that a person evaluates visual and numerical characteristics in parallel, but they do not have enough cognitive control to inhibit irrelevant visual parameters. This interpretation assumes that visual parameters are pre-attentive and evaluated automatically, making them difficult to suppress. It is possible that incongruent tasks require more resources to suppress the effect of irrelevant visual parameters. In this case, the resources of attention may not be sufficient to estimate the quantity more accurately, and the NRE increases.

Previously, several scholars postulated that the processing of quantity might be flexible, adapting to the demands of the environment [41]. Expanding this suggestion, our results revealed that the interplay between non-numerical visual and numerical features depends on many factors, including the format of presentation and numerical ratio. Our findings demonstrated that the estimation of numerical information might be accurate enough, even if the estimation of visual cues is impeded. Conversely, when the estimation of numerosity became noisier, the role of non-numerical information increased, which was reflected in the increase in the congruency effect for the large proportion. At the same time, the congruency effect was significant in the separate format only and its increase in the large proportion condition was salient only in this format. Thus, we can suggest that the processing of non-numerical cues can be involved in processing numerosity but this is not mandatory. In general, the obtained results are in line with the ANS theory.

### 4.1. Limitations

Our study has several limitations. First, we estimated the NRE as the difference between small and large proportions. In the “large proportion” condition, the ratio between the two compared arrays varied from 0.72 to 0.77. Meanwhile, in other studies, several types of proportions were included, including larger proportions such as 0.85 and higher [46]. Some authors have suggested that the relationship between ANS acuity and the numerical ratio has a nonlinear effect [26]. Therefore, it is possible that the obtained findings are related to a narrow range of numerical ratios. In future studies, it is worth including a higher value of numerical proportions.

Moreover, despite our efforts to balance the numerical size between the “small proportion” and “large proportion” conditions, there were instances in the “large proportion” conditions where the numerical size was larger (e.g., 31) than in the “small proportion” condition. Consequently, we did not completely separate the numerical ratio effect (NRE) and the size effect in this experimental design. Future studies should aim to simultaneously control for both NRE and the size effect.

Another limitation stems from the characteristics of the sample and the online testing procedure. Our sample consisted of university students, and they demonstrated a high level of accuracy in the nonsymbolic comparison task. It is possible that for individuals with less precise number sense, the congruency effect and the NRE would have different relationships. In addition, testing was conducted in an online format, and numerous participants did not finish the test. We can suggest that those participants who completed the test had a higher level of motivation and/or a higher level of number sense. Therefore, the obtained results might be related to participants with a high level of motivation and competence.

One more limitation may be a consequence of the test design. As we included an additional condition to test the difference between the comparison of triangles and circles, in the whole test, we had more congruent trials than incongruent ones, and they were presented in random order. Previously, it was shown that in several conflict monitoring tasks, such as the Stroop test (which also contains congruent and incongruent trials), the size of the congruency effect depended on the sequence of items, and it was reduced if an incongruent trial was followed by another incongruent trial [47]. These results were extended in a recent study by Viarouge, Lee, and Borst (2023), who demonstrated that the congruency effect in a nonsymbolic comparison task was reduced when the conflicting dimension was the same in the preceding incongruent trial (for example, trials were incongruent on the same visual cue) [48]. It was also shown that in the nonsymbolic comparison test, the congruency effect was greater when trials of different types were mixed [49]. We did not control for the sequence of items, so the sequence might affect the congruency effect, but we cannot estimate this effect.

However, despite these limitations, the obtained results confirmed that the mechanisms of numerosity estimation include processing both visual and numerical features and that the interaction between these two processes varies depending on the format of presentation and numerical proportion.

### 4.2. Conclusions

In this study, we extended previous findings regarding the variability of the congruency effect in the nonsymbolic comparison test. This study is the first to investigate the interaction between the congruency effect and NRE in different formats of the nonsymbolic comparison test. The results revealed that the congruency effect was not significant in the intermixed format for both small and large numerical proportions. At the same time, the congruency effect had the highest values in the separated format for a large numerical proportion and was lower for a small numerical proportion.

In summary, our study may indicate that the ability to process numerical information is based on the estimation of both visual and quantity features, and that the interaction between these two processes varies depending on the format of presentation. It may be suggested that evaluating visual parameters may serve as an auxiliary process when the estimation of numerosity becomes noisier.

## Figures and Tables

**Figure 1 behavsci-13-00983-f001:**
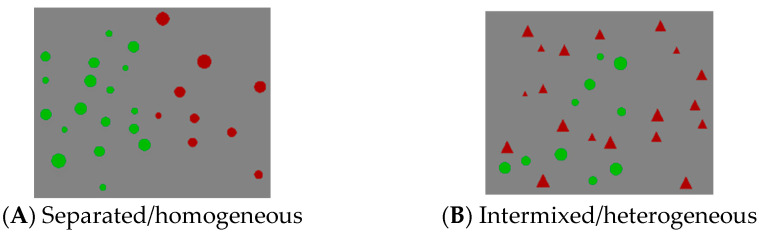
Examples of stimuli for two formats of the nonsymbolic comparison test.

**Figure 2 behavsci-13-00983-f002:**
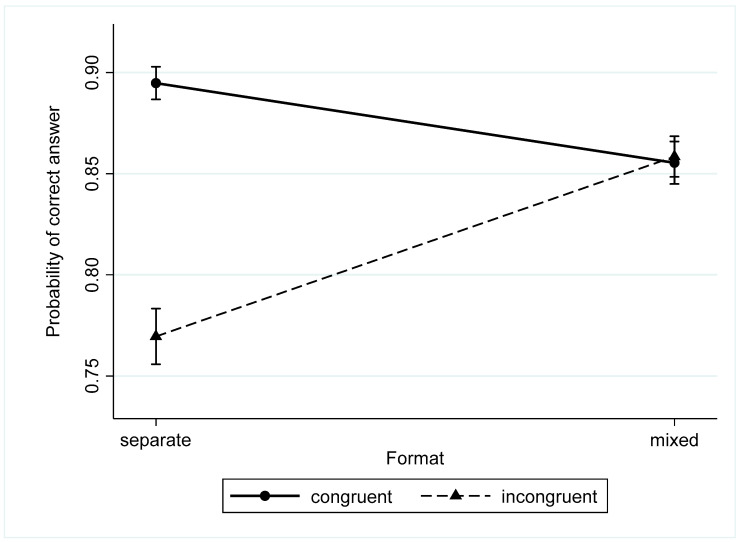
The probability of correct answers for congruent and incongruent trials in different formats.

**Figure 3 behavsci-13-00983-f003:**
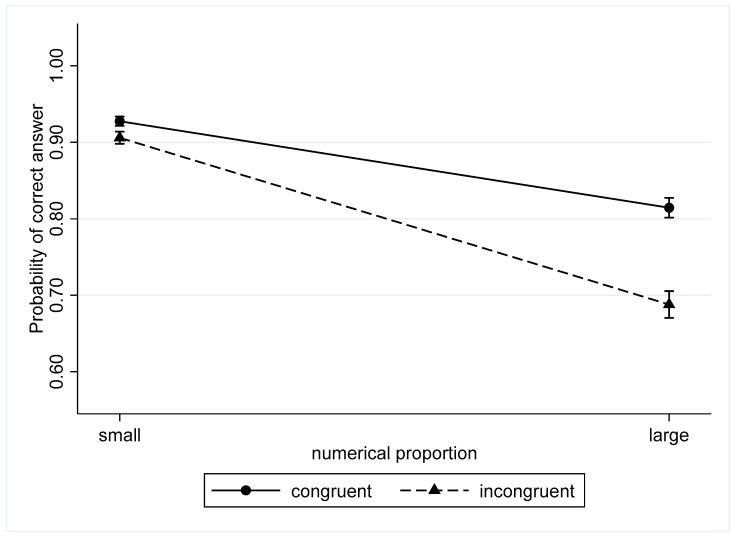
The probability of correct answers for congruent and incongruent trials in small and large proportions.

**Figure 4 behavsci-13-00983-f004:**
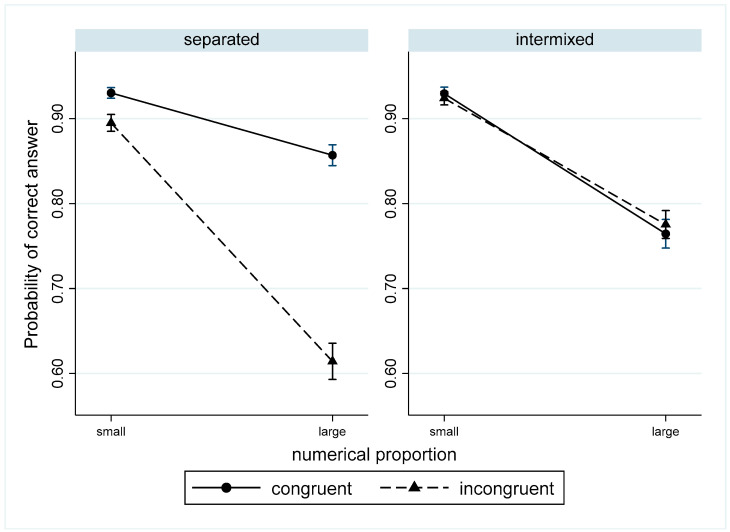
The probability of correct answers for congruent and incongruent trials in different formats for small and large proportions.

**Table 1 behavsci-13-00983-t001:** Description of test conditions.

Condition	Figures	Format	Numerical Proportion	Congruency	Number of Figures (Red:Green)
1	Circles	Separated	Small	Congruent	9:18; 9:19; 16:8; 19:10
2	Circles	Separated	Small	Incongruent	9:18; 9:19; 16:8; 19:10
3	Circles and triangles	Intermixed	Small	Congruent	9:18; 9:19; 16:8; 19:10
4	Circles and triangles	Intermixed	Small	Incongruent	9:18; 9:19; 16:8; 19:10
5	Circles	Separated	Large	Congruent	9:12; 13:18; 13:10; 16:12
6	Circles	Separated	Large	Incongruent	9:12; 13:18; 13:10; 16:12
7	Circles and triangles	Intermixed	Large	Congruent	9:12; 13:18; 13:10; 16:12
8	Circles and triangles	Intermixed	Large	Incongruent	9:12; 13:18; 13:10; 16:12

**Table 2 behavsci-13-00983-t002:** Descriptive statistics in the nonsymbolic comparison test.

Conditions	Mean	SD	95% CI	Range
Whole test	0.83	0.09	0.82; 0.84	0.51–0.98
Small proportion	0.91	0.10	0.90; 0.92	0.52–1.00
Large proportion	0.74	0.09	0.73; 0.75	0.43–0.95
Intermixed format	0.83	0.09	0.82; 0.84	0.47–0.98
Separated format	0.83	0.09	0.82; 0.84	0.50–0.98
Congruent	0.87	0.09	0.86; 0.88	0.50–0.98
Incongruent	0.79	0.10	0.78; 0.80	0.41–0.99

**Table 3 behavsci-13-00983-t003:** Proportion of correct answers in the nonsymbolic comparison test in each condition.

Numerical Proportion	Separate Format		Intermixed Format	
	Congruent	Incongruent	Congruent	Incongruent
Small	0.92 (0.01)	0.88 (0.01)	0.92 (0.01)	0.92 (0.01)
Large	0.84 (0.01)	0.61 (0.01)	0.75 (0.01)	0.76 (0.01)
Both	0.89 (0.01)	0.74 (0.01)	0.83 (0.01)	0.83 (0.01)

**Table 4 behavsci-13-00983-t004:** GLMM results for the probability of correct answers in the nonsymbolic comparison test.

Variables	Baseline Model	Model 1	Model 2	Model 3
	B (s.e.)	B (s.e.)	B (s.e.)	B (s.e.)
Fixed effects				
Intercept	1.72 *** (0.04)	2.62 *** (0.04)	2.70 *** (0.05)	2.59 *** (0.05)
Intermix.format		0.18 *** (0.02)	−0.31 *** (0.04)	−0.01 (0.05)
Incongruent		−0.51 *** (0.02)	−0.70 *** (0.04)	−0.45 *** (0.05)
Large prop.		−1.32 *** (0.02)	−1.02 *** (0.04)	−0.80 *** (0.05)
Interactions				
Format*Incongr.			1.02 *** (0.05)	0.37 *** (0.08)
Format*Propor.			−0.10 (0.05)	−0.60 *** (0.07)
Propor.*Incongr.			−0.44 *** (0.05)	−0.88 *** (0.07)
Propor.*Incongr.*Format				1.02 *** (0.10)
Random effects				
Between-individual variance	0.34	0.40	0.41	0.42
Log-likelihood	−27,068.55	−25,074.56	−24,812.72	−24,759.24
LR test (df)		3987.98 *** (3)	523.70 *** (3)	106.95 *** (1)
ICC	0.09			

*** *p* < 0.001.

**Table 5 behavsci-13-00983-t005:** The congruency effect and the NRE in separated and intermixed formats.

Effects	Separated Format	Intermixed Format
	B (Log Odds) (s.e.)	B (Log Odds) (s.e.)
The congruency effect	−0.90 *** (0.03)	0.13 *** (0.04)
Numerical ratio effect	−1.22 *** (0.03)	−1.32 *** (0.04)

*** *p* < 0.001.

**Table 6 behavsci-13-00983-t006:** The congruency effect in different conditions.

Conditions	Small Proportion	Large Proportion
	B (Log Odds) (s.e.)	B (Log Odds) (s.e.)
Intermixed format	−0.08 (0.06)	0.06 (0.04)
Separated format	−0.45 *** (0.05)	−1.33 *** (0.04)

*** *p* < 0.001.

**Table 7 behavsci-13-00983-t007:** The NRE under different conditions.

Conditions	Congruent Trials	Incongruent Trials
	B (s.e.)	B (s.e.)
Intermixed format	−1.40 *** (0.05)	−1.26 *** (0.05)
Separated format	−0.80 *** (0.05)	−1.68 *** (0.05)

*** *p* < 0.001.

## Data Availability

Data are available on reasonable request through correspondence with the authors.

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
