# Peer review of "The Interaction between Congruency and Numerical Ratio Effects in the Nonsymbolic Comparison Test"

_behavsci, 2023, doi:10.3390/bs13120983_

Round 1
Reviewer 1 Report
Comments and Suggestions for Authors
The authors report a study aimed to estimate how the congruency effect varied depending on the format of presentation and numerical proportion. A mixed-effects models was applied to a sample of 290 students and estimated the interactions between congruency, format of presentation, and numerical proportion. On my own reading of the manuscript I certainly think your paper has the potential to make a contribution, but I concur with the reviewers that while it is interesting it requires further work.
1. The abstract should provide a clear summary of the paper's main findings and conclusions. It currently focuses more on introducing the topic of the study rather than summarizing the results.
2. The introduction should clearly state the purpose of the study and what research question it aims to answer. It currently just mentions the general topic of approximate number sense and the nonsymbolic comparison task without clearly stating the specific focus of the study.
3. Could the authors please provide a justification of their sample size.
4. It was mentioned that” There are two concurrent theories regarding the ANS and its relationships with visual cues.” (Line 335). The author should compare and analyze the two theories, as well as which theory supports the experimental results.
Comments on the Quality of English LanguageModerate editing of English language required
Author Response
Reviewer 1
Dear reviewer,
We are grateful for opportunity to correct our manuscript. Below we answered on each comment.
Comments and Suggestions for Authors
The authors report a study aimed to estimate how the congruency effect varied depending on the format of presentation and numerical proportion. A mixed-effects models was applied to a sample of 290 students and estimated the interactions between congruency, format of presentation, and numerical proportion. On my own reading of the manuscript I certainly think your paper has the potential to make a contribution, but I concur with the reviewers that while it is interesting it requires further work.
- The abstract should provide a clear summary of the paper's main findings and conclusions. It currently focuses more on introducing the topic of the study rather than summarizing the results.
Answer:
Thank you for your opinion. We re-write abstract in order it reflects better main conclusions of our research. Current version of abstract: “The nonsymbolic comparison task is used to investigate the precision of the Approximate Number Sense, the ability to process discrete numerosity without counting. There is an ongoing debate regarding the extent to which the ANS is influenced by the processing of non-numerical visual cues. To address this question, we assessed the congruency effect in a non-symbolic comparison task, examining its variability based on different stimulus presentation formats and numerical ratios between comparing arrays. Additionally, we explored the variability of the numerical ratio effect in relation to the format and congruency. Utilizing generalized linear mixed-effects models with a sample of 290 students (89% female, mean age 19.33 years), we estimated the congruency effect and numerical ratio effect for both separated and intermixed formats of stimulus presentation.
The findings indicated that the congruency effect was more pronounced for a larger numerical proportion and in the separated format. Three-way interactions between format, congruency and numerical ratio revealed that the congruency effect increased in larger numerical proportion, but this pattern was observed only in the separated format. In the intermixed format the congruency effect was insignificant for both types of numerical proportions. Notably, the numerical ratio effect varied for congruent and incongruent trials in different formats. Results may suggest that the processing of visual non-numerical parameters may be crucial when numerosity processing becomes noisier, specifically when numerical proportion becomes larger. The implications of these findings for refining the ANS theory are discussed.
- The introduction should clearly state the purpose of the study and what research question it aims to answer. It currently just mentions the general topic of approximate number sense and the nonsymbolic comparison task without clearly stating the specific focus of the study.
Answer:
We are sorry that introduction was not clear enough for you. We added some information and identified the main points and aims of our study.
- Could the authors please provide a justification of their sample size.
Answer:
We have assumed that effect size for the congruency effect was 0.3 (based on previous studies it may vary) and the power was 0.9. According to this point, the required sample size was 234. For calculating sample size package statsmodels for Python was used.
- It was mentioned that” There are two concurrent theories regarding the ANS and its relationships with visual cues.” (Line 335). The author should compare and analyze the two theories, as well as which theory supports the experimental results.
Answer:
Thank you for this comment. We discussed these theories in Introduction. In corrected version of Manuscript we clearly identified two alternate theories and highlighted differences between them (p. 1 -2 of corrected version). We have also mentioned these theories in Discussion section regarding our findings (p.12).

Reviewer 2 Report
Comments and Suggestions for Authors
The authors present an interesting paper on the interaction between congruence and numerical relatedness.
I consider that the theoretical framework is well founded with abundant bibliographical references, although most of these references are around 10-15 years old. Without being a specialist in the subject, are there no more current references?
The authors should go a little deeper into the definition of Approximate Number Sense, as they present a very limited vision of the term.
In relation to the method, I consider that the work is well designed and the analysis of the results has been rigorous.
In relation to the discussion, I consider that the first two paragraphs of the section, rather than a discussion, it seems that we have returned to the theoretical framework of the article. It does not make sense to show a summary of the theoretical framework, but rather to contrast the results of this work with those of other authors.
In the third paragraph of the discussion, the authors express a judgment based on an opinion (lines 344-345). I consider that this is not acceptable in this type of work, it is more convenient that they back up their assertion with the data of the work.
Only from line 354 onwards do we start to see a real discussion of the results of the study with other research.
There are some errors in the form of explicit citations, for example in lines 96, 112, and 209. There are also errors in the final list of references. For example, parts of the journal names are in bold, and others are not when they should not be.
Author Response
Reviewer 2
Dear reviewer,
We are grateful for opportunity to correct our manuscript. Below we answered on each comment.
Comments and Suggestions for Authors
The authors present an interesting paper on the interaction between congruence and numerical relatedness.
I consider that the theoretical framework is well founded with abundant bibliographical references, although most of these references are around 10-15 years old. Without being a specialist in the subject, are there no more current references?
Answer:
Thank you for this point. We used many paper which were written earlier because they were relevant to our topic. Besides, many of them are supposed to be “classic” papers for topic on numerosity processing (like Dehaene, 2001, 2003 or Halberda, Mazzocco, Feigenson, 2008). Meanwhile, we don’t agree that the most references in our paper are old. We calculated the proportion of references from different papers. We have 48 references on papers, 25% of them were published last 5 years (in 2018 – 2023) years, 25% were published in 2015 – 2017 years, 25% were published 2010 – 2014, and 25% were published since 2001 till 2009. In summary, in our manuscript we considered as enough old and classic papers and novel publications.
The authors should go a little deeper into the definition of Approximate Number Sense, as they present a very limited vision of the term.
Answer:
Thank you for this point. We are agree that there exist different views on Approximate number sense and its definitions. We are focused on more detailed definition and did not include more broader process such as subitizing or texture-density, or nonsymbolic arithmetic and so on. We did this because the topic of interrelations between processing of non-numerical cues and numerosity is mostly discussed regarding to this “narrow” definition.
We have also assumed that NRE and congruency effect is related to the ANS, so discussion of these effects are related to the ANS characteristics.
In relation to the discussion, I consider that the first two paragraphs of the section, rather than a discussion, it seems that we have returned to the theoretical framework of the article. It does not make sense to show a summary of the theoretical framework, but rather to contrast the results of this work with those of other authors.
Answer:
Thank you for this point. We re-write discussion and shortened these paragraphs.
In the third paragraph of the discussion, the authors express a judgment based on an opinion (lines 344-345). I consider that this is not acceptable in this type of work, it is more convenient that they back up their assertion with the data of the work. Only from line 354 onwards do we start to see a real discussion of the results of the study with other research.
Answer:
Thank you for this notification. We believed that discussion section is the section where authors might not only discussed results but express judgments too, especially since our judgment was based on previous studies (e.g. Leibovich-Raveh et al., 2018) and our results. However, we have changed this paragraph: “Previously, it has been postulated that the processing of quantity might be flexible, adapting to the demands of the environment [41]. Expanding this suggestion, our results revealed that the interplay between non-numerical visual and numerical features depends on many factors, including the format of presentation and numerical ratio. Our findings demonstrated that the estimation of numerical information might be accurate enough, even if the estima-tion of visual cues is impeded. And conversely, when the estimation of numerosity became noisier, the role of non-numerical information increased that reflected in the increase of the congruency effect. At the same time, assuming that the congruency effect increased in large proportion for separate format only, we can suggest that processing of non-numerical cues is not mandatory for processing numerosity. In general, the obtained results are in line with the AMS theory”.
There are some errors in the form of explicit citations, for example in lines 96, 112, and 209. There are also errors in the final list of references. For example, parts of the journal names are in bold, and others are not when they should not be.
Answer:
Thank you for this notification. We used Zotero for formatting references in Style of Behavior Science. In Word document all titles of journals are in italic. We supposed that the formatting might be changed during uploading files into management system. We have also corrected the reference style in text.

Round 2
Reviewer 2 Report
Comments and Suggestions for Authors
Thank you very much for addressing all the suggestions for improvement made in the first review.
Author Response
Thank you for your valuable comments and reviewer.